

# Enhanced genome editing in human iPSCs with CRISPR-CAS9 by co-targeting *ATP1a1*

Jui-Tung Liu,  James L. Corbett,  James A. Heslop and  Stephen A. Duncan

Department of Regenerative Medicine and Cell Biology, Medical University of South Carolina, Charleston, SC, United States of America

## ABSTRACT

Genome editing in human induced pluripotent stem cells (iPSCs) provides the potential for disease modeling and cell therapy. By generating iPSCs with specific mutations, researchers can differentiate the modified cells to their lineage of interest for further investigation. However, the low efficiency of targeting in iPSCs has hampered the application of genome editing. In this study we used a CRISPR-Cas9 system that introduces a specific point substitution into the sequence of the $Na^+/K^+$-ATPase subunit ATP1A1. The introduced mutation confers resistance to cardiac glycosides, which can then be used to select successfully targeted cells. Using this system, we introduced different formats of donor DNA for homology-directed repair (HDR), including single-strand DNAs, double-strand DNAs, and plasmid donors. We achieved a 35-fold increase in HDR when using plasmid donor with a 400 bp repair template. We further co-targeted *ATP1A1* and a second locus of interest to determine the enrichment of mutagenesis after cardiac glycoside selection. Through this approach, INDEL rate was increased after cardiac glycoside treatment, while HDR enrichment was only observed at certain loci. Collectively, these results suggest that a plasmid donor with a 400 bp repair template is an optimal donor DNA for targeted substitution and co-targeting *ATP1A1* with the second locus enriches for mutagenesis events through cardiac glycoside selection in human iPSCs.

## INTRODUCTION

Precise genome editing technology provides researchers with a powerful tool to investigate the function of genes in nearly any species. By using programmable nucleases to cut the genome, researchers can mutate or correct the genes of interest for further investigation. Although there are still concerns about safety issues, such technologies may pave the way to develop new therapies for cancer treatment or genetic diseases (*Cornu, Mussolino & Cathomen, 2017*; *Fu et al., 2013*; *Hsu et al., 2013*; *Ihry et al., 2018*; *Pattanayak et al., 2013*).

The most widely used technologies to introduce double-strand breaks (DSBs) are zinc-finger nucleases (ZFNs), transcription activator-like effector nucleases (TALENs), and the RNA-guided cluster regularly interspaced short palindromic repeats-Cas9 (CRISPR-Cas9) system (*Joung & Sander, 2013*; *Sander & Joung, 2014*; *Urnov et al., 2010*). ZFNs and

Corresponding author
Stephen A. Duncan,
duncanst@musc.edu

TALENs link endonuclease catalytic domains to DNA-binding proteins to generate DSBs. Alternatively, the CRISPR-Cas9 system uses small guide RNAs that are paired with target DNA to induce DSBs by Cas9. The CRISPR-Cas9 appears to be more flexible, efficient, and easier to design, when compared to the other two systems (*Gaj, Gersbach & Barbas 3rd, 2013*).

Generating DSBs at the target site activates two primary endogenous DNA repair mechanisms in mammalian cells (*Valerie & Povirk, 2003*). The most frequent repair pathway is non-homologous end joining (NHEJ), which is based on the direct ligation of two DNA ends without the need for a homologous template (*Kanaar, Hoeijmakers & Van Gent, 1998*). This error-prone mechanism can create an insertion or deletion of bases in the genome (INDEL), which may disrupt gene expression by introducing frameshift mutations. This methodology has been an important tool for deleting genes of interest across many species (*Bapteste & Philippe, 2002*; *Belinky, Cohen & Huchon, 2010*; *Luan et al., 2013*). Homology-directed repair (HDR) is a more precise mechanism that is based on DNA recombination between genomic DNA and the homologous template (*Johnson & Jasin, 2001*). HDR allows the genomic sequences to be modified based on the introduced repair template. Utilizing HDR machinery for genome editing provides advantages for genome engineering due to the accuracy and predictable nature of the mutation. Numerous studies have applied this genome editing strategy to therapeutic approaches (*Deverman et al., 2018*; *Shim et al., 2017*; *WareJoncas et al., 2018*).

Applying genome-editing tools to human induced-pluripotent stem cells (iPSCs) has provided new directions for regenerative medicine and *in vitro* disease modeling (*Robinton & Daley, 2012*). Several studies have successfully generated cell lines to recapitulate genetic diseases using the CRISPR/Cas9 system (*Ben Jehuda, Shemer & Binah, 2018*). Despite the advantages, the efficiency of creating mutations via NHEJ or HDR remains relatively low in human iPSCs (*Mali et al., 2013*; *Wang et al., 2013*). Single-stranded DNA oligonucleotides (ssODNs) have been used as repair templates to efficiently introduce single-nucleotide mutations, and double strand DNA donor plasmids are used for fragment insertion via HDR (*Chen et al., 2011*). However, HDR rates vary depending on cell type and status (*Saleh-Gohari & Helleday, 2004*). Researchers have proposed strategies to enhance the success rate of genome editing, including cell cycle synchronization, introduction of selection markers, and pre-treatment with small molecule NHEJ inhibitors (*Chu et al., 2015*; *Guo et al., 2018*; *Yu et al., 2015*). Recently, researchers applied Cas9 ribonucleoproteins in combination with AAV-mediated repair template delivery to increase integration frequency (*Martin et al., 2019*). Although the efficiency is improved from the original technique, the timing of drug treatment and administration of CRISPR-Cas9 are difficult to control due to the cell cycle variation between cell lines.

Cardiac glycosides have been used to treat heart failure by targeting ATP1A1, a subunit of $Na^+/K^+$-ATPase (*McDonough et al., 2002*; *Smith, 1984*). With exposure to relatively high concentrations of such drugs, cell viability is reduced via the accumulation of intracellular $Ca^{2+}$ levels (*Belusa et al., 2002*; *Lin et al., 2017*). The binding site of cardiac glycosides on ATP1A1 has been identified, and N-terminal amino-acid substitution of ATP1A1 encoded by exon4 (Q118R and N129D) is sufficient to confer drug resistance by preventing the
binding of the cardiac glycosides (*Treschow et al., 2007*). In 2017, *Agudelo et al. (2017)* used "co-CRISPR" methods to target exon 4 of *ATP1A1* concurrently with a second locus of interest in established human cell lines. By selecting *ATP1A1* mutations using ouabain, the percentage of cells harboring an INDEL at the second locus increased (*Agudelo et al., 2017*). However, the co-CRISPR strategy was not applied to human iPSCs and in the cell lines that were tested the HDR rate was not significantly improved.

Here, we report the use of a CRISPR-Cas9 editing system that targets intron 4 of *ATP1A1* in human iPSCs. By introducing a repair template to substitute the two N-terminal amino acids Q118 and N129, we specifically created an HDR-directed mutation that confers resistance to cardiac glycosides. We then applied different types of repair templates to investigate the efficiency of HDR. Furthermore, by co-targeting with the second locus of interest, we examined the efficiency of HDR-directed mutation after selection with the cardiac glycoside digoxin.

## MATERIALS & METHODS

### Construction of CRISPR/Cas9 plasmids and donor templates

CRISPR guide RNAs were designed obeying the NGG PAM sequence rule. The construction of CRISPR plasmids followed the protocol established by *Ran et al. (2013)*. To generate CRISPR construct targeting *ATP1A1*, a guide sequence (5′-GAGTTCTGTAATTCAGCATA-3′) was cloned into PX459 pSpCas9(BB)-2A-Puro plasmid following the fast digest protocol (*Ran et al., 2013*). The gRNAs targeting the other locus were cloned in to PX459 via the same procedure and the sequences of gRNAs were listed as below: (DGUOK exon1: 5′-CGAAGGCTCTCCATCGAAGG-3′; DGUOK exon4: 5′-CATCGAGTGGCATATCTATC-3′; POLG exon10: 5′-ATGCAGGGTCGTCTAGCCGG-3′; GATA6 exon4: 5′-TTATGGCGCAGAAACGCCG-3′; RYR2B exon50: 5′-TGACAGGGTCTATGGGATTG-3′). Single-stranded oligodeoxynucleotides (ssODNs) were obtained as Ultramer® DNA oligonucleotides (IDT, IL, US). Designed double-stranded DNAs were generated through GeneArt™ synthesis with the addition of restriction enzyme cutting sequences at both ends (Thermo Fisher Scientific, MA, US). To construct donor plasmids, dsDNAs from GeneArt™ synthesis were cut with restriction enzymes and cloned into pBlueScript II KS(+) (Addgene, MA, US). The sequences from all constructs were sequenced and verified before proceeding with the CRISPR experiment. Donor templates for each targeted locus are listed in Table S1.

### Cell lines, cell culture and electroporation

Two human induced pluripotent stem cell lines, K3 hiPSCs (*Si-Tayeb et al., 2010*) and SV20 hiPSCs (*Yang et al., 2015*), which were characterized in previous studies, were used for this study. Provenance of all cell lines and use in these experiments were approved by the MUSC Stem Cell Research Oversight Committee protocol #8. Cells were cultured in mTeSR feeder-free defined medium (*Ludwig et al., 2006*) supplemented with 4 ng/mL zebrafish basic fibroblast growth factor. All iPSCs were maintained as colonies on E-cadherin-IgG fusion protein matrix at 37 °C with 4% $O_2$/5% $CO_2$ (*Nagaoka & Duncan, 2010*). Electroporation of plasmids and ssODNs used electroporator ECM630 (BTX, MA, US) following the
manufacturer's instruction. Briefly, iPSCs were expanded until 80% confluence on one 100 mm dish to give approximately $4 \times 10^7$ cells and were harvested in small clumps using 0.05% EDTA. Cells were mixed with purified DNAs in 4 mm electroporation cuvette and immediately electroporated at 250 volts/200 $\Omega$/700$\mu$F. Transfected cells were cultured onto 100 mm dish coated with Matrigel (2 mg/ml) (Invitrogen, MA, US) in mTeSR medium supplemented with 4 ng/mL zebrafish basic fibroblast growth factor and 10 $\mu$M of Y27632 (STEMCELL Technologies, VBC, CA) for 24 h. For *ATP1A1* co-targeting, cells were transfected with PX459 plasmids containing guide sequence of *ATP1A1* and second gene of interest, along with linearized donor plasmids following the electroporation protocol above. After 24 h of Y27632 treatment, transfected cells were selected by culturing with 1$\mu$g/mL of puromycin for 48 h. The survived cells were then recovered for an extra 48 h with the normal mTeSR medium before digoxin selection. To obtain digoxin-resistant cells, 1 $\mu$M of digoxin (Sigma, MO, US) was added on to the medium for 72 h with medium changes performed on a daily basis. The surviving cells were harvested for genomic DNA extraction.

### RFLP and TIDE(R) analysis for genome modification

The genomic DNA of iPSCs was harvested using QuickExtract™ DNA extraction solution following the manufacturer's instruction (Epicentre, Illumina, WI, US). Each single cell colony or the whole population of cells in a 100 mm dish were harvested in 50 $\mu$L or 1 mL of QuickExtract reagent, respectively. Out-out PCR amplification was performed using Herculase II fusion DNA polymerase (Agilent, CA, US) with the following conditions: (95 °C for 5 min; 40×: 95 °C for 30 s, 60 °C for 30 s, 72 °C for 30 s; 72 °C for 5 min). PCR primers were located at least 100 bp outside of donor templates region to prevent amplifying donor template. Primers for each targeted locus are listed in Table S2. For RFLP analysis, PCR amplicons were purified via QIAquick PCR Purification Kit (Qiagen, MD, US), and 400 ng of purified amplicons were digested EcoRI or NruI, respectively. Digested DNA was then separated on a 2% agarose gel stained with ethidium bromide. For TIDE(R) analysis, purified PCR products were sequenced by Retrogen (CA, US), and the sequence signal of the experimental group was compared with wild-type or donor template as negative or reference control, respectively. TIDE(R) data analysis is described by *Brinkman et al. (2018)*, and a free web tool is available at http://tide.nki.nl.

### Immunostaining and cell viability assay

Wild-type and $ATP1A1^{Q118R/N129D}$ iPSCs were treated with 15.625, 31.25, 62.5, 125, 250, 500, and 1,000 nM of digoxin and ouabain for 72 h in a 96-well plate coated with Matrigel (2 mg/ml) (Invitrogen, MA, US). A cell viability assay was performed using CellTiter-Glo® luminescent cell viability assay kit following the manufacturer's instructions (Promega, WI, US). For nucleus immunostaining, cultured cells were fixed with 4% paraformaldehyde for 20 min following 0.5% Triton X-100 treatment for 15 min. Cells were blocked with 3% bovine serum albumin in PBS for 30 min and incubated with DAPI (1 $\mu$g/ml) at room temperature for 30 min. Fluorescence intensity was assessed with ZOE™ Fluorescent Cell Imager (Bio-rad, CA, US). Experimental and control groups were processed identically.

## Statistical analysis

Results generated by cell viability assay, survival colonies counting, and TIDE(R) analysis were expressed as mean SD. Data were analyzed by ANOVA followed by Tukey's or by Student's $t$-test, as appropriate. Statistical significance was achieved at $P < .05$.

# RESULTS

## Designing CRISPR-Cas9 system to edit ATP1a1 locus

Cardiac glycosides inhibit $Na^+/K^{+-}$ATPase pump activity by binding to the ATP1A1 subunit. The blockage leads to the accumulation of $Ca^{2+}$ and causes cell apoptosis (*Riganti et al., 2011*). Treschow et al. identified two amino acids in exon 4, Q118 and N129, that are responsible for the binding of cardiac glycosides to the $Na^+/K^{+-}$ATPase (*Treschow et al., 2007*). With Q118R and N129D substitutions, cells became resistant to high doses of ouabain without affecting the function of the $Na^+/K^{+-}$ATPase (*Treschow et al., 2007*). We, therefore, proposed to facilitate endogenous selection of targeted events in iPSCs using the CRISPR-Cas9 system to introduce the Q118 and N129 mutations into ATP1A1. We designed a guide RNA targeting intron 4 of *ATP1A1*, with the PAM site twenty base pairs downstream of exon 4 (Fig. 1A). In order to generate the amino-acid substitution allele, we designed a 153 single-stranded oligodeoxynucleotides (ssODN) with the sequences that replace glutamine with arginine at position 118 and asparagine with aspartic acid at position 129 (Fig. 1A). After transfecting both sp-Cas9-gRNA (px459-*ATP1A1*-intron4) and donor template into the cells and treating these cells with a high dose of digoxin, we collected the surviving clones for genotyping (Fig. 1B). Using primers outside the targeted homology arms to perform PCR (out-out PCR) and restriction fragment length polymorphism (RFLP) analysis, we observed that all the cells that were resistant to digoxin or ouabain were successfully repaired through HDR, and both heterozygous and homozygous mutations were identified (Fig. 2A). The genotype was confirmed by genomic sequencing. In order to determine the optimum dosage of digoxin and ouabain, we performed a dose–response assay of both drugs on wild-type and *ATP1A1* homozygous mutant iPSCs. We effectively eliminated all wild-type iPSCs at 1 μM after 72 h of exposure. In contrast to the parental iPSCs, homozygous ATP1A1 mutant iPSCs survived in the presence of either drug and were indistinguishable from untreated cells (Figs. 2B–2F). These results demonstrate that cardiac glycosides can be used to effectively identify HDR-driven mutations at the endogenous *ATP1A1* locus.

## Comparing the HDR efficiency while using different repair templates

Although single-stranded oligodeoxynucleotides (ssODNs) have commonly been used as repair donor templates, the efficiency is generally low when using iPSCs (*Radecke et al., 2010*; *Yoshimi et al., 2016*). Double-strand DNAs with long homologous arms have been shown to improve the efficiency of introducing changes through genome editing in HEK293T cells (*Song & Stieger, 2017*). We therefore proposed to compare the efficiency of HDR-mediated mutagenesis between ssODNs and dsDNA at the *ATP1A1* locus using endogenous selection. We co-transfected human iPSCs with px459-*ATP1a1*-intron4 and donor template presented in the following formats: ssODNs, dsDNAs, or dsDNAs cloned

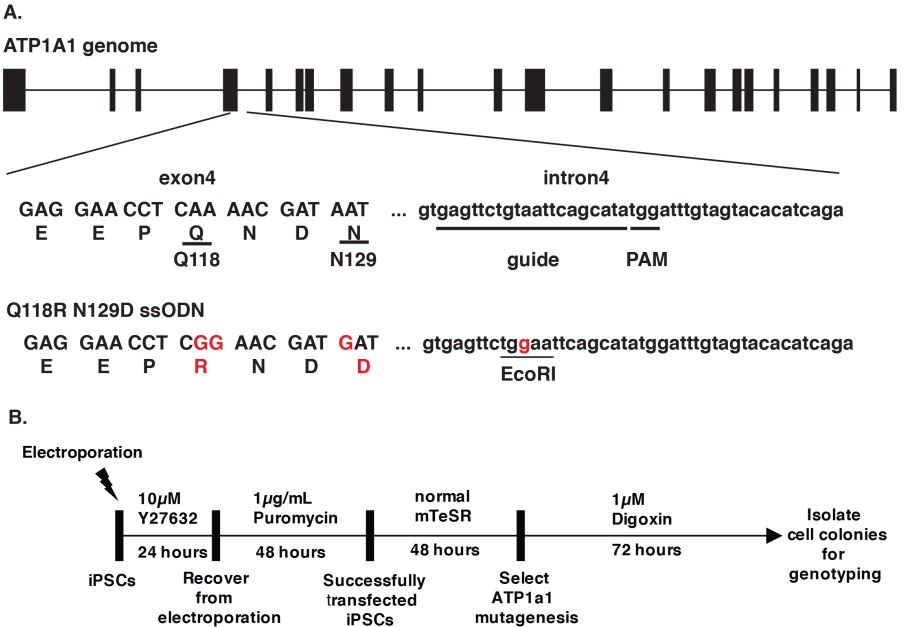

**Figure 1  Design of CRISPR/Cas9 targeting ATP1A1.** (A) The design of *ATP1A1* guide RNA and single-stranded oligonucleotide (ssODN) donor. Guide RNA was designed to target intron 4 of the *ATP1A1* gene. The expected cut site is 3–4′ nucleotides upstream of the Protospacer Adjacent Motif (PAM) sequence and 20 nucleotides downstream of the exon 4 region. The donor template was designed to introduce nucleotide substitutions that result in the replacement of Glutamine with Arginine (Q118R) and Asparagine with Aspartic acid (N129D). The EcoRI cut site was introduced to allow identification of positive clones. (B) The timeline of performing gene-editing on human iPSCs. Human iPSCs were transfected with a CRISPR/Cas9 plasmid targeting *ATP1A1* intron 4 and a repair template through electroporation. Cells were treated with 10 μM Y27632 for 24 h following 48 h of puromycin exposure to select transfected iPSCs. Cells were maintained for an extra 48 h until colonies were observed and before administering digoxin for 72 h. Surviving cells were isolated and genotype determined by sequence analyses and PCR.

into a plasmid. HDR-positive clones were then selected using a 72-hour exposure of 1 μM digoxin (Fig. 3A). As shown in Fig. 3B, up to two positive clones were observed when co-transfected with 150 base-pair (bp) template either as ssODN, dsDNA, or linearized plasmid (Fig. 3B). Strikingly, when cells were co-transfected with 400 bp template either as dsDNA or linearized plasmid, the surviving colonies increased 11-fold and 35-fold, respectively when compared to transfection using 150 bp donors. These results indicate that double-stranded repair templates with ≥400 bp length have a higher efficiency of generating HDR-positive cells. Moreover, assembling the dsDNA into a plasmid vector improved the effectiveness by a further ∼3-fold (Fig. 3B).

## Co-editing an independent locus by selecting for mutation of ATP1A1 in iPSCs

To determine if selecting digoxin-resistant cells by modifying *ATP1a1* locus increases the efficiency of HDR-driven events at a second locus of interest, we used the CRISPR-Cas9 system to co-target *ATP1A1* and deoxyguanosine kinase (*DGUOK*) gene in iPSCs. The *DGUOK* genetic alteration that we chose to introduce results in a loss of function

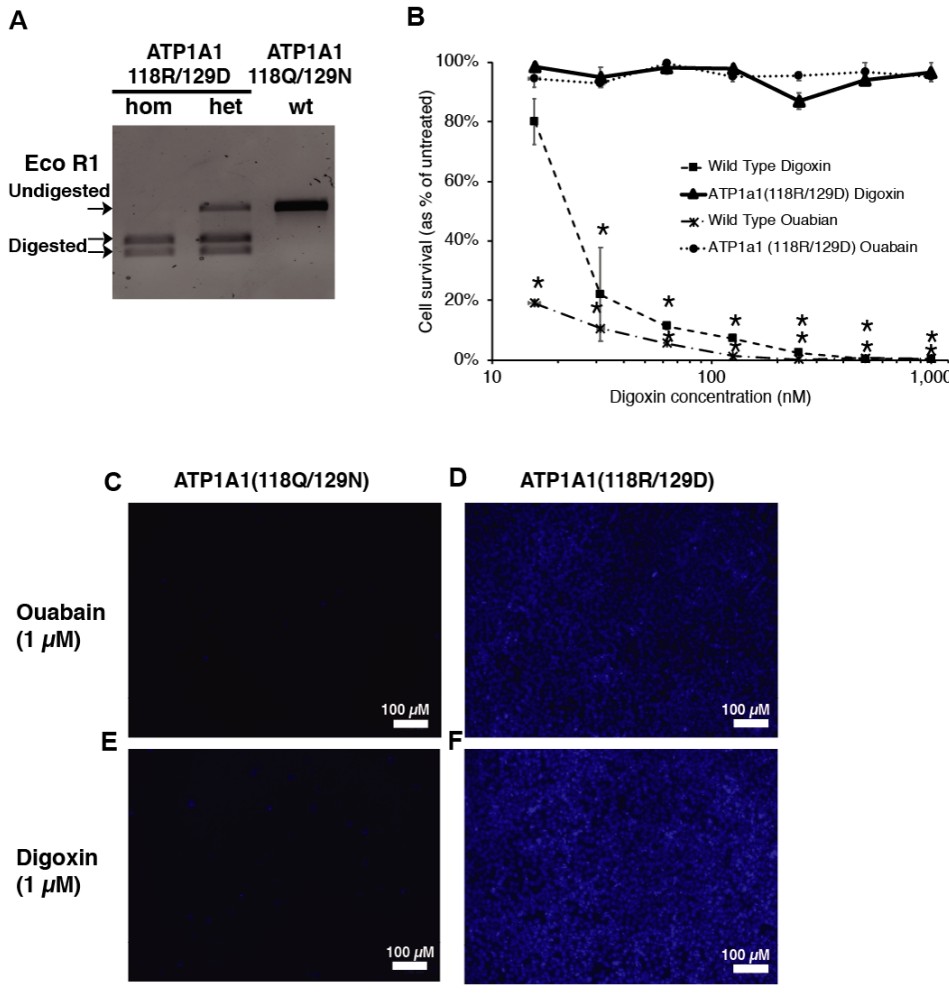

**Figure 2** **Introduction of ATP1A1$^{Q118R/N129D}$ in human iPSCs confers resistance to cardiac glycosides.**
(A) RFLP analysis of ATP1A1 mutation. Transfected iPSCs were selected through 1 µM of digoxin exposure and genomic DNA from each individual colony was extracted for out-out PCR reaction. PCR amplicons were digested by EcoRI to identify homozygous and heterozygous mutations (for raw data see Fig. S1). (B) Wild-type iPSCs and homozygous ATP1A1 mutants were exposed to 15.625, 31.25, 62.5, 125, 250, 500, and 1000 nM of digoxin and ouabain for 72 h. Cell viability was determined by luminescence assay (B, for raw data see Data S1) and was confirmed by DAPI staining (C–F) (for uncropped images see Figs. S2—S5). The experiment was conducted in triplicate ($n = 3$) and data are shown as mean ± SD. Statistical differences were determined by ANOVA followed by Tukey's test (*$p < .01$). (Wild Type Oubain: $n = 3$, $df = 8$, $F = 5731.75$, $p = 4.85 \times 10^{-29}$; Wild Type Digoxin: $n = 3$, $df = 8$, $F = 37.33$, $p = 1.17 \times 10^{-9}$).

mutation, DGUOK Q170X, which is observed in patients with Mitochondrial DNA Depletion Syndrome 3 (Hepatocerebral type) (MTDPS3); the patients with this syndrome commonly die from liver failure and suffer from progressive dysfunction of neural and muscular tissues (*Mandel et al., 2001*). Two different Human iPSC lines (SV20 and K3) (*Jing et al., 2018*; *Si-Tayeb et al., 2010*; *Yang et al., 2015*) were co-transfected with CRISPR plasmids targeting *ATP1A1* and *DGUOK* respectively, along with the linearized donor plasmids designed to introduce point mutations to each locus (Fig. 4A). Transfected cells

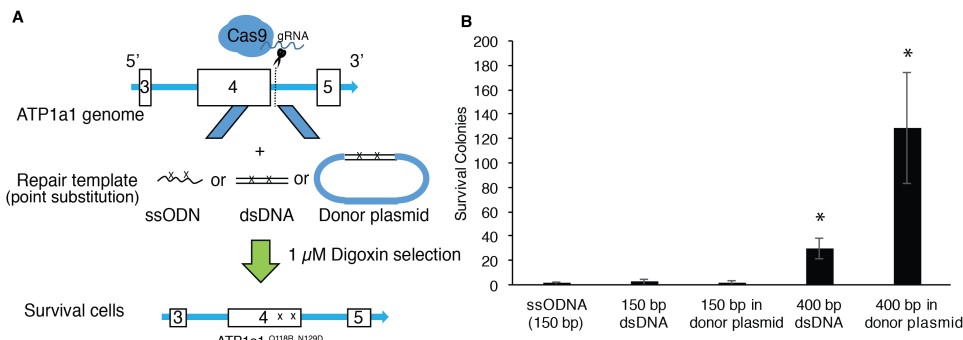

**Figure 3** Provision of long HA dsDNA donor template during gene editing increases HDR-mediated mutagenesis. (A) Diagram of approach used to edit *ATP1A1* using different donor templates. $1 \times 10^7$ of human iPSCs were transfected with PX459 containing guide RNAs which target intron 4 via electroporation. Cells were also transfected with 0.01 nmole of ssODN, dsDNA, and linearized donor plasmid, respectively. Seven days after electroporation, cells were treated with 1 μM of digoxin for 72 h. (B) The number of surviving iPSC colonies for each group was counted. The experiment was conducted in triplicate ($n = 3$). Data are shown as mean ± SD. Statistical differences were determined by ANOVA followed by Tukey's test (*$p < .05$). ($n = 3$, $df = 4$, $F = 21.17$, $p = 7.19 \times 10^{-5}$).

were then cultured until colonies were observed. At this point, cells were treated with or without 1 μM of digoxin for 72 h. In order to calculate the efficiency of HDR, genomic DNA was collected and PCR using primers outside the targeted homology arms of *DGUOK* exon4 (out-out PCR) was performed to identify restriction fragment length polymorphisms (RFLP). Amplicons were also sequenced and the frequency of indels and targeted mutations determined using TIDE/TIDER analyses (*Brinkman et al., 2018*). Strikingly, we observed a significant increase in HDR-mediated introduction of mutations after digoxin selection (Fig. 4B). TIDER analysis confirmed a two to three-fold increase in HDR-driven events from 4% to 13% in K3 and 8.3% to 19% in SV20 iPSCs (Fig. 4C). Moreover, the INDEL rate also dramatically increased from 11.7% to 39% in K3 and 24.3% to 46.9% in SV20 iPSCs (Fig. 4C). The results demonstrate that after digoxin treatment to select cells with Q118 and N129 mutations in ATP1A1, both INDEL and HDR-driven targeting increased substantially at the second locus.

## HDR-driven event varies between genomic location

To determine if the co-selection for the ATP1A1[Q118R/N129D] increases efficiency across different loci, we tested the same approach at other genes of interest, including *RYR2*, *GATA6*, and *POLG*, all of which are located on different chromosomes. We also targeted the *DGUOK* gene at a different independent locus. Each gene was targeted individually or along with *ATP1A1*. Transfected cells with or without digoxin selection were collected for out-out PCR, and INDEL-HDR rates were determined using TIDE/TIDER analysis. As shown in Table 1, the INDEL rates without digoxin selection ranged from 1.3% to 45.6%, which is likely due to different inherent CRISPR guide targeting efficiencies. After digoxin selection, INDEL efficiency significantly increased among all targeted loci (Table 1). HDR rates remained low without digoxin selection, ranging from 0 to 4.18%. After digoxin treatment, the cells showed an increase of HDR at both *DGUOK* regions; however, this

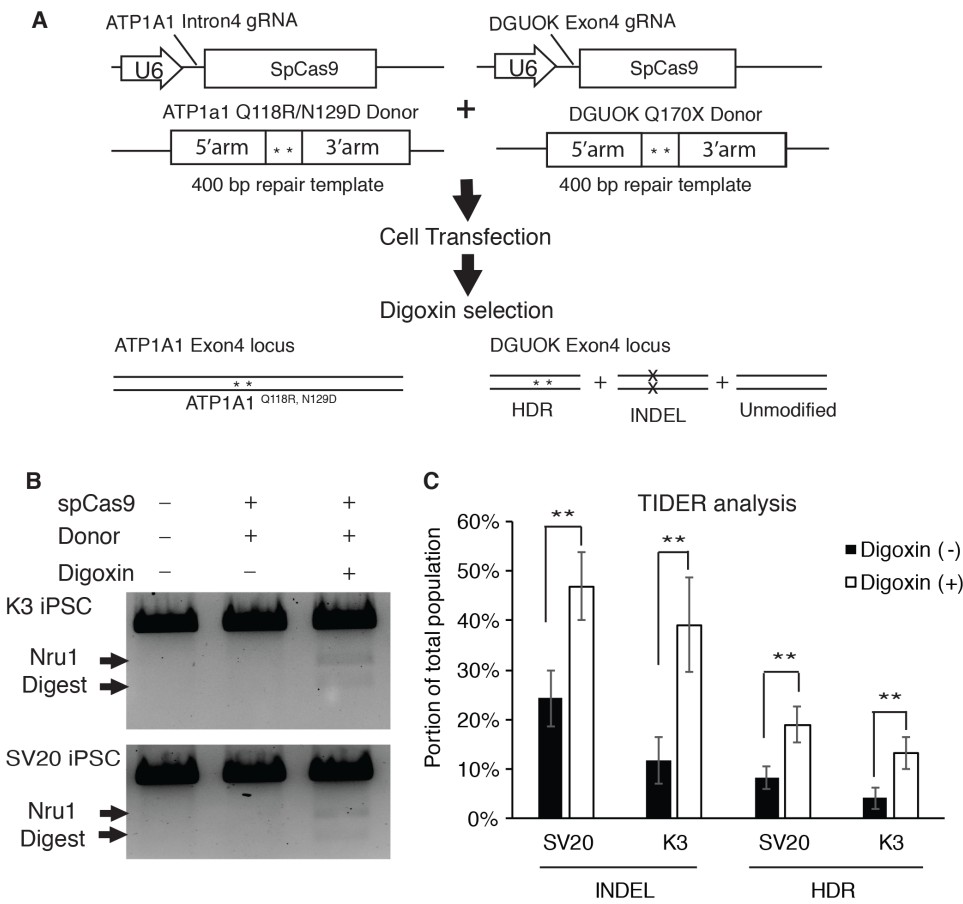

**Figure 4  Selecting for mutation of *ATP1A1* by digoxin enriches HDR and INDEL-mediated mutagenesis at the *DGUOK* gene.** (A) Diagram of CRISPR co-targeting appraoch. Two CRISPR plasmids PX459 containing guide RNAs (*ATP1A1* Intron 4 and *DGUOK* exon 4), and two linearized repair templates were transfected into two different iPSC lines (K3 and SV20). The DGUOK repair template introduced a NruI restriction enzyme cutting sequence to facilitate RFLP analysis. Cell selection was demonstrated in Fig. 1B. (B) Cells with or without culture in 1 μM digoxin were collected and genomic DNA was extracted from the whole population. RFLP analysis was performed to determine the level of enrichment of HDR-driven mutagenesis (for uncropped image see Fig. S6). (C) Out-out PCR amplicons from (B) were sequenced and subjected to TIDER analysis (for raw data see Data S2). The genome editing experiment was performed in five replicates ($n = 5$). Data are shown as mean $\pm$ SD. Statistical differences between each two means (Digoxin-treated V.S. non-treated) were determined by student $T$-test ($^*p < .05$, $^{**}p < .01$) (SV20 Indel: $p = 5.1 \times 10^{-4}$, K3 Indel: $p = 3.34 \times 10^{-4}$, SV20 HDR: $p = 1.33 \times 10^{-3}$, K3 HDR: $p = 3.64 \times 10^{-4}$).

increase was not observed at the other targeted loci (Table 1) presumably due to the excessively low efficiency of HDR at these specific sites. In summary, the results indicate that the *ATP1A1* co-targeting approach successfully selected targeted cells, and although HDR efficiency increases, the rate remains variable between loci.

## DISCUSSION

In the present study, we used CRISPR-Cas9 to edit the *ATP1A1* locus in human iPSCs. The introduced mutations inhibited the binding of cardiac glycosides, thus conferring

**Table 1  The population of HDR and INDEL-mediated mutagenesis after co-targeting *ATP1A1* and a second locus of interest.**  Five different loci were targeted by CRISPR-Cas9 in human iPSCs along with two linear repair templates (*ATP1A1* and a second locus of interest). After transfection cells were cultured in the presence or absence of digoxin selection, cells were collected and sequenced for TIDE(R) analysis. The percent population of INDELs and HDR events are shown.

| Gene | Chromosome | Indel | | HDR | |
|---|---|---|---|---|---|
| | | Digoxin (−) | Digoxin (+) | Digoxin (−) | Digoxin (+) |
| DGUOK_EX1 | 2 | 16.53% | 43.10% | 0% | 10.85% |
| DGUOK_EX4 | 2 | 11.76% | 39.03% | 4.18% | 13.23% |
| RYR2_EX50 | 1 | 12.40% | 43.53% | 1.35% | 3.13% |
| GATA6_EX4 | 18 | 45.60% | 81.80% | 0.40% | 0% |
| PolG_EX10 | 15 | 1.38% | 25.33% | 0% | 0% |

resistance to high dosage of this class of drug. More importantly, we were able to obtain genomic-modified clones that were specifically repaired by HDR after ouabain and digoxin selection. We showed that using 400 bp dsDNA as the repair template increased HDR efficiency compared to 150 bp ssODNs, while delivering the template in a linearized plasmid further improved HDR rates. Moreover, by co-targeting *ATP1A1* with a second locus of interest, we were able to increase both INDEL and HDR selection efficiency in two different iPSC lines. Although HDR-driven events varied between genomic loci, the overall efficiency of selecting genetically-modified clones increased. The optimization of the genome-editing strategy in human iPSCs provided by this study will pave the way for future disease modeling and gene therapy research.

Optimizing repair templates to obtain increased HDR-driven mutations after double-strand breaks has been well studied (*Baker et al., 2017*; *Song & Stieger, 2017*; *Zhang et al., 2017*). Due to difficulties in generating ssODNs longer than 200 bp, researchers have used single strand DNA with 30–70 base homology arms (HA) as donor template for small edits (*Ran et al., 2013*; *Richardson et al., 2016*; *Yang et al., 2013*). The dsDNAs are generally used as the repair templates for fragment insertions (*Byrne et al., 2015*; *Urnov et al., 2005*). In human iPSCs, the optimal HA length is reported to be ∼2 kb (*Byrne et al., 2015*). Our study tested the efficiency of ssODN, dsDNA and linearized plasmid for CRISPR-based genomic editing through HDR. We found that homology arms that are less than 200 bp showed no significant difference between all three forms of donor DNA. We initially reasoned that this result might be due to low HDR efficiency in iPSCs, as other studies showed that less than 2% of DSB repair was mediated through the HDR pathway in stem cells (*Yang et al., 2013*). Strikingly, when we extended the length of dsDNA to 400 bp, there was a dramatic increase in HDR efficiency. Our observations are similar to a previous report showing that when the length of donor DNA was over 300 bp (150 bp HA), the efficiency of HDR-driven events increased (*Zhang et al., 2017*). Moreover, when using linearized plasmids containing the repair template in the center region, the number of positive clones was significantly higher than non-plasmid dsDNA fragments. This observation may be due to DNA degradation after transfecting into the cells, while linearized plasmids have higher stability because of the protection inferred by the backbone sequence. The results are also consistent with

another study showing that using PCR products as repair templates are not as efficient as plasmid donors (*Song & Stieger, 2017*). Although applying long homology arm dsDNA as repair template is significantly improved targeting, in this study it was only applied to *ATP1A1*. Whether this finding is suitable across multiple loci remains to be rigorously determined; however, application of plasmids with large homology domains at *DGUOK* and *RYR2* also enhanced our ability to retrieve mutations at these sites.

The two major mechanisms to repair double-strand DNA breaks are NHEJ and HDR. Previous studies have shown that HDR appears to be favored when cells are in the S-G2 phase of the cell cycle (*Lieber, 2010*; *Sonoda et al., 2006*). Based on our observations, cells repaired by HDR on *ATP1A1* after DSBs had both high activities of INDEL and HDR on the second locus, *DGUOK*, while other co-targeted genes didn't show significant improvement on HDR. Given that we did not attempt cell cycle synchronization, the observed differences in efficiencies are likely independent of the cell-cycle. Our data are also consistent with Agudelo's study that after ouabain selection, INDEL rate significantly increased on the second targeted locus (*Agudelo et al., 2017*). Nevertheless, it is still unknown why selecting cells with HDR-driven mechanism on *ATP1A1* enriched the overall mutagenesis population on the second locus. Recently, chromatin accessibility has been found to be associated with CRISPR-Cas9 efficiency in zebrafish. The authors of that study indicated that the open chromatin is more likely to be targeted by CRISPR-Cas9 (*Uusi-Makela et al., 2018*). In the future, it would be interesting to investigate whether co-targeting with *ATP1A1* and treating with digoxin preferentially selects cells that have greater chromatin accessibility, and are therefore more susceptible to genome editing.

Variations within the $Na^+/K^+$-ATPase that confer resistance to cardiac glycoside have been found to occur naturally across species, including insects, metazoans, and rats (*El-Mallakh, Brar & Yeruva, 2019*; *Perne et al., 2009*; *Ujvari et al., 2015*). Q118R and N129D mutations in *ATP1A1* have been rigorously examined and found to have no impact on $Na^+/K^+$-ATPase activity (*Price, Rice & Lingrel, 1990*). Instead the Q118R and N129D variations simply prevent the stable association of the glycoside with extracellular domain of the $Na^+/K^+$-ATPase. Therefore, ATP*1A1 Q118R;N129D* to confer resistance to glycoside toxicity should have no impact on cellular function. However, as is the case with all drug selection, there is a remote chance that conferring resistance to glycosides could have unintended consequences that should be considered (*Askari, 2019*).

We have described one use for the endogenous selection approach which is to optimize conditions needed for efficient targeting. However, one could envision multiple uses for the same approach. For example, selection of glycoside resistance could be used to efficiently generate allelic variations in iPSCs that would facilitate disease modeling, *ex vivo* gene-editing therapy, and drug discovery (*Corbett & Duncan, 2019*; *Lee et al., 2020*). Also, having an additional endogenous selectable allele will be helpful when there is a need to generate iPSC lines either with multiple variations within the same cells, revertant alleles, or rescue constructs.

Editing genomic DNA in iPSCs is challenging, not only due to relatively low HDR/INDEL rates, but also the colony-style culture methods which limit the capacity for single-cell sorting selection. Scientists have reported several strategies to address this challenge,

including modifying medium, pre-treating with small molecules, and overexpressing anti-apoptotic genes such as *BCL2* to increase cell viability during genome editing (*Byrne & Church, 2015*; *Chen & Pruett-Miller, 2018*; *Gonzalez et al., 2014*; *Li et al., 2018*). To improve the selection efficiency, researchers have also showed a significant increase on HDR/INDEL rate by introducing excisable markers such as PiggyBac or Cre/lox system; however, both these methods significantly increase the time required to generate the mutations of interest (*Wang et al., 2017*; *Zhu et al., 2015*).

## CONCLUSIONS

We present a study using CRISPR-Cas9 to edit the *ATP1A1* locus in human iPSCs and determining that introducing 400 bp dsDNA repair template increased HDR efficiency compared to 150 bp ssODNs. Moreover, we observed that by co-targeting *ATP1A1* with a second locus of interest, the INDEL and HDR selection efficiency were improved in two different iPSC lines. Our described approach, using a one-step co-targeting strategy with a longer dsDNA repair template, will shorten the period of selection process and increase both HDR/INDEL rate. The Na$^+$/K$^+$ channel is thought to be critical for maintaining cell osmolarity, thus the disruption of ion-exchange ability results in cell death (*Pierre & Xie, 2006*). Since *ATP1A1* mutation described here only affects the binding of glycosides and does not affect the function of Na$^+$/K$^+$ pump, the approach has the advantage of being of marker-free selection that is suitable for clinical application.

## ACKNOWLEDGEMENTS

We thank Dr. Morad and Dr. Yamaguchi for kindly providing CRISPR/Cas9 plasmid and donor templates targeting *RYR2*. We thank Drs. Pournasr, Jing, and Furio, for guidance and manuscript editing.

### Funding

This work is supported by grants from the United States National Institutes of Health (R01DK102716, R01DK119728 and P20GM130457). The funders had no role in study design, data collection and analysis, decision to publish, or preparation of the manuscript.

### Grant Disclosures

The following grant information was disclosed by the authors:
United States National Institutes of Health: R01DK102716, R01DK119728, P20GM130457.

### Competing Interests

Stephen A. Duncan is founder and CEO of Gruthan Bioscience, LLC.

### Author Contributions

- Jui-Tung Liu conceived and designed the experiments, performed the experiments, analyzed the data, prepared figures and/or tables, authored or reviewed drafts of the paper, and approved the final draft.

- James L. Corbett and James A. Heslop performed the experiments, authored or reviewed drafts of the paper, and approved the final draft.
- Stephen A. Duncan conceived and designed the experiments, analyzed the data, prepared figures and/or tables, authored or reviewed drafts of the paper, and approved the final draft.

## DNA Deposition

The following information was supplied regarding the deposition of DNA sequences:

The sequence of the ATP1A1 gene is available at Ensembl: ENSG00000163399.

## Data Availability

The raw measurements are available in the Supplemental Files.

## Supplemental Information

Supplemental information for this article can be found online at http://dx.doi.org/10.7717/peerj.9060#supplemental-information.

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
