# Peer review of "Enhanced genome editing in human iPSCs with CRISPR-CAS9 by co-targeting ATP1a1"

_PeerJ, doi:10.7717/peerj.9060_

## Round 0.1 · original submission · Minor Revisions

Your paper has been seen by two expert reviewers and overall, they are very supportive of publication and have only minors comments. I would therefore like to invite you to respond to the referees’ comments in a form of a revised version of this manuscript.

Reviewer 1 ·

Basic reporting

This manuscript from Liu et al. aims to investigate CRISPR HDR efficiency in the generation of series of point mutations in iPSCs. To achieve this the authors transfected Px459 plasmid with various HDR templates (single stranded DNA, double stranded DNA…) and double selected the mutations via Puromycin and Digoxin selection. Overall the authors found that CRISPR HDR efficiency firstly increases with the use of longer homology arms and double stranded DNA and sequence/gene dependent.

Overall this is a very well crafted manuscript. The manuscript is well written, well structured, well referenced.

I have however noted a few spelling mistakes in the manuscript and the Figure legends. Please proofread again the manuscript

Experimental design

The hypotheses are well stated, the introduction and background shows well the context and the manuscript well structured. The methods section describes well the experiments. The results presented are well ascertained and rigorously interpreted. The methods are well described and in sufficient details.

The only comment I have for the authors is the following:

L125: 40 mm electroporation cuvette seems not correct. Do the authors refer to instead a 4 mm cuvette?

Validity of the findings

This is overall a nice study, well conducted and well performed. the data provided are statistically sound and the conclusions well stated. I do have however one additional minor comments to the authors.

My comment relates to the comparison between different donor templates (L270-290) and Figure 3B. The authors investigated the homology arms length and their types (ssODN, dsDNA…) and found the number of survival colonies increased for a donor plasmid with long homology arms. The length of the homology arms, symmetry and type of donor to improve HDR efficiency has been largely explored since the use of the Zinc Finger nucleases. While the results are sound, my main criticism is this investigation was performed on one locus only. I therefore believe the authors should add a note in the discussion section better outlining the limitations of this finding.

·

Basic reporting

This is an interesting study to introduce a second “silent selectable mutation” in ATP1a1. This allows to select iPSC cells where HDR editing can occur, on-target editing of the desired site increased by 35 fold.

The article is clear and the supporting data seem valid.

Experimental design

The question is well defined and testable. The methods are appropriate

Validity of the findings

The experiments are sound. It extends the findings of other groups that us ATPa1 mutations as a selection marker.

Additional comments

This is a clever technique, but the disadvantage for disease modeling, since there is a new mutation in ATPa1 could still have some other unknown consequences. For disease modeling most groups want a true isogenic line. Also with improvements in iPSC transfection, and Cas9 RNP the vast majority of edits can be isolated fairly easily.

It would be helpful to the discussion, if the authors could provide specific use-cases for needing this type of selection with ATPa1.

---

## Round 0.2 · accepted · Accept

The authors carefully took into account the reviewer's comments. The current version of the article is suitable for publication in PeerJ.